# 3D Cross-Pseudo Supervision (3D-CPS): A semi-supervised nnU-Net architecture for abdominal organ segmentation

Yongzhi Huang[1,2,#,], Hanwen Zhang[1,2,#,],
Yan Yan[1,2,], and Haseeb Hassan[1,2,*,]

[1] College of Big Data and Internet, Shenzhen Technology University, Shenzhen, 518188, China
[2] College of Applied Sciences, Shenzhen University, Shenzhen, 518060, China

**Abstract.** Large curated datasets are necessary, but annotating medical images is a time-consuming, laborious, and expensive process. Therefore, recent supervised methods are focusing on utilizing a large amount of unlabeled data. However, to do so, is a challenging task. To address this problem, we propose a new 3D Cross-Pseudo Supervision (3D-CPS) method, a semi-supervised network architecture based on nnU-Net with the Cross-Pseudo Supervision method. We design a new nnU-Net based preprocessing. In addition, we set the semi-supervised loss weights to expand linearity with each epoch to prevent the model from low-quality pseudo-labels in the early training process. Our proposed method achieves an average dice similarity coefficient (DSC) of 0.881 and an average normalized surface distance (NSD) of 0.913 on the 2022-MICCAI-FLARE validation set (20 cases).

**Keywords:** Abdominal organ segmentation · Semi-supervised learning · 3D Cross-Pseudo Supervision.

## 1 Introduction

With the advancement of medical technology, more and more medical imaging devices are being used in clinical diagnosis [1]. Nowadays, the analysis of medical images by physicians to diagnose any possible disease has become the basis of clinical diagnosis [2]. Therefore, image segmentation is the first step in the process of many clinical diagnoses and plays a key role in the quantitative analysis of medical images [3]. With the rapid development of deep learning in recent years, various U-Net-based network models have achieved excellent results in different medical image segmentation competitions [4]. Among them,

---

[1] # These two authors contributed equally to this work.
[2] Corresponding author: Haseeb Hassan(haseeb@sztu.edu.cn).

the nnU-Net performs significantly well with its excellent adaption to different datasets and achieves excellent segmentation results under fully supervised medical datasets using a plain U-Net network [5]. However, the manual annotation for segmentation is time-consuming and needs expert experience, especially in medical images. Thus, recent trends have focused on automatically using a large amount of unlabeled medical images. For instance, a semi-supervised approach has been introduced to construct a teacher-student model with dropout or uncertainty to generate stable segmentations automatically over unlabeled images [6,7]. Likewise, Li et al. and Luo et al. proposed a model for different tasks, and then this model can be trained with consistency losses on unlabeled data [8,9].Above all, these inspired semi-supervised-based approaches still have the following challenges.

1. Data preprocessing pipelines and algorithms that rely on labeled data can not be applied directly to unlabeled data.
2. Semi-supervised strategies for processing data are often used only in binary tasks, while multi-class 3D medical tasks will take more time and memory consumption in the data processing.
3. To deal with the imbalance problem in the number of labeled and unlabeled data, an appropriate training strategy is required to learn from different losses.

To address the above challenges, we design a semi-supervised method based on nnU-Net by improving the Cross Pseudo-label algorithm. The proposed method applies the preprocessing of nnU-Net to unlabeled data and further adapts the Cross Pseudo-label algorithm(CPS) [10] to 3D CT images. The proposed method is validated on Fast and Low-resource semi-supervised Abdominal oRgan sEgmentation in CT challenge (2022-MICCAI-FLARE). The main contributions of our work are summarized as follows.

1. Modified the nnU-Net architecture for semi-supervised tasks.
2. Adopted a semi-supervised strategy called Cross-Pseudo Supervision for 3D medical images.

## 2   Method

### 2.1   Preprocessing

**Training**  In the training stage, we use the same preprocessing pipeline generated from the heuristic rules as in the nnU-Net on labeled data, including intensity transformation, spatial transformation, and data augmentation. The only difference is that when analyzing the distribution of CT intensity, nnU-Net analyzes the intensity of the foreground so that the label is required. However, we obtain the intensity distribution of unlabeled data by unifying a statistical method for calculating the intensity of the whole image by collecting global pixels information instead of foreground pixels.

**Inference**  We follow the nnU-Net in the inference stage: adopting the TTA inference, setting the step size of the sliding window 0.7 and using "normal mode" in nnU-Net.

## 2.2  Proposed method

**Semi-supervised learning on nnU-Net architecture**  Our proposed framework is shown in Fig. 1, which is adopted from the nnU-Net. For segmentation tasks on a specific dataset, the pipeline of the nnU-Net framework mainly includes the following steps:

1. Collect and analyze the data information, and generate rule-based parameters using its heuristic rules.
2. Train the network based on fixed parameters and rule-based parameters.
3. Perform ensemble selection and post-processing methods based on empirical parameters.

Our proposed framework keeps the pipeline of the nnU-Net but modifies some of their components such as for data fingerprint collection. We replaced the distribution of foreground pixels with global pixels to analyze the intensity distribution of unlabeled data, as mentioned in section 2.1. Secondly, we doubled the U-Net network architecture and optimizer generated from the heuristic rules of the nnU-Net. The sibling networks have the same architecture but with different initialization. Correspondingly, the two sibling networks have their optimizer to update the weights. Then, the proposed semi-supervised method trains the network in the training stage, which will be discussed in detail in the next section. Since the model ensembling module is of no use for comparing the plain 2D

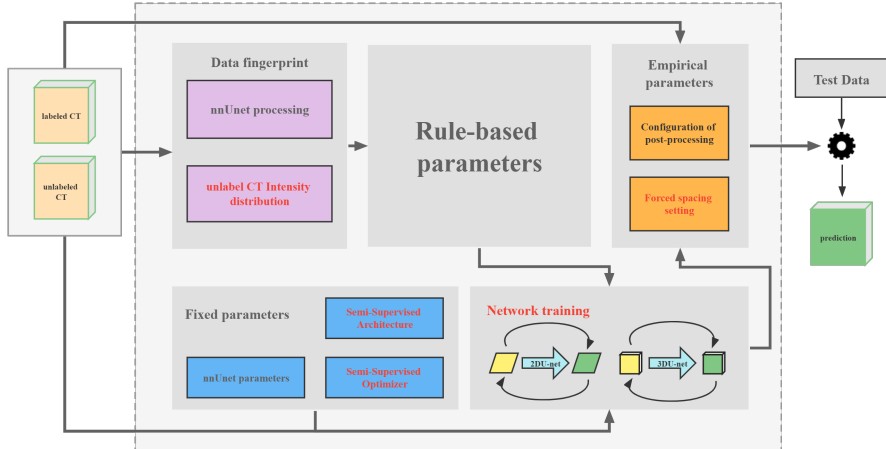

**Fig. 1.** The 3D CPS framework: a semi-supervised learning framework based on nnU-Net architecture. The proposed framework entirely follows nnU-Net and we modified their respective blocks including intensity distribution analysis, network architecture, optimizer, training policy, and inference preprocessing in the framework which is marked as red. The forced spacing settings is only used in experiments of the final test.

and 3D U-Net architectures with or without the CPS, so it is removed from the proposed framework. Additionally, in the inference stage, the enforced spacing settings is used before preprocessing to improve the segmentation efficiency.

**Proposed semi-supervised method** The proposed semi-supervised method is mainly derived from CPS. Though the CPS is a 2D semi-supervised semantic segmentation method, this method can be adopted for 3D CT images. A semi-supervised semantic segmentation task aims to learn a segmentation network by exploring labeled and unlabeled data. For instance, given a set $D_l$ of labeled images and a set $D_u$ of unlabeled images. The proposed method consists of two parallel segmentation networks $T_1$ and $T_2$, with the same network architecture but were initialized with different weights $\theta_1$ and $\theta_2$. The input $X$ can be either a 2D patch with the shape of [C, H, W] or a 3D patch with the shape of [C, S, H, W].

The semi-supervised training strategy is shown in Fig. 2, where the input $X$ is preprocessed with the same pipeline as the nnU-Net. Then, the default data augmentation in nnU-Net is applied for the input X. $T_1(X)$ and $T_2(X)$ represent the predicted one-hot confidence map from the two parallel $T_1$ and $T_2$ networks, respectively. $Y_1(X)$ and $Y_2(X)$ represent the predicted one-hot label map generated from $T_1(X)$ and $T_2(X)$, respectively.

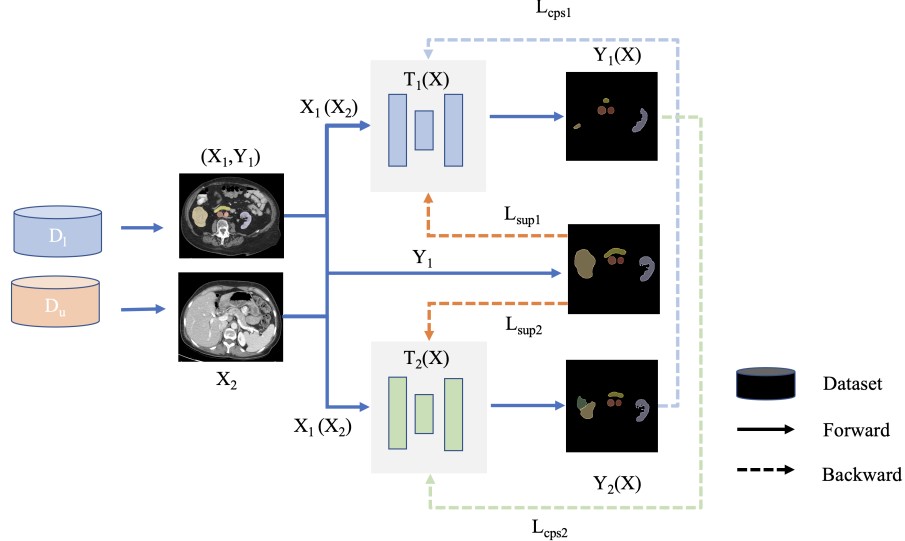

**Fig. 2.** Modified semi-supervised architecture from cross-pseudo supervision(CPS). Labeled data $(X_1, Y_1)$ and unlabeled data $X_2$ are sampled randomly as input for both networks $T_1(\cdot)$ and $T_2(\cdot)$. For any input X, the output is a pair of branches, i.e., $T_1(X)$ and $T_2(X)$. Labeled data and unlabeled data are calculated based on cross-pseudo supervision loss. Moreover, the labeled data also requires calculation of supervision loss.

Furthermore, our training objective contains two losses: supervision loss $L_{sup}$ and cross-pseudo supervision loss $L_{cps}$. The supervision loss $L_{sup}$ can only be calculated for the labeled data and is given as follows.

$$L^l_{sup} = l_{sup}(T_1(x), Y) + l_{sup}(T_2(x), Y) \tag{1}$$

where $Y$ is the ground truth, and $l_{sup}$ is dice and cross-entropy loss configured by default in nnU-Net, which is proved to be robust in medical image segmentation tasks [11].

The cross-pseudo supervision loss $L_{cps}$ is defined as $L_{cps} = L^l_{cps} + L^u_{cps}$, which is combined with the CPS loss on labeled dataset $L^l_{cps}$ and on unlabeled dataset $L^u_{cps}$. The two parallel networks view the output of the other one as own pseudo-labels and the $L_{cps}$ can be calculated as $L_{sup}$. The cross-pseudo supervision loss on the unlabeled data is written as:

$$L^u_{cps} = l_{cps}(T_1(x), Y_2) + l_{cps}(T_2(x), Y_1) \tag{2}$$

where $l_{cps}$ is dice and cross-entropy loss like $l_{sup}$ in our experiments. Given Eq. 2, the definition of CPS loss on labeled data is the same as the loss on unlabeled data. The overall training objective is given as:

$$L = L_{sup} + \lambda L_{cps} \tag{3}$$

where $\lambda$ is a hyper-parameter that needs to be set in advance, determining the weight of cross supervision loss. We set the $\lambda$ to increase linearity with epoch from 0 to 0.5, and keep that fixed after the specific epoch instead of the fixed value as in CPS.

**Network architecture** Our network is a U-Net [12] like Encoder-Decoder architecture, and its backbone is based on residual blocks [13]. Following the configurations of nnU-Net, two different U-Net architectures (2D and 3D) are generated from heuristic rules of nnU-Net. Their detailed structures are provided in Fig. 3 and Fig. 4, respectively.

The 2D CPS U-Net can only take one slice as input. During the training and inference stage, each slice needs to be resized to 512× 512. The network is an 8-layers U-Net, and the encoding process downsamples images to 4× 4. The implementation of residual blocks is as follows: conv-instnorm-ReLU, and each encoder contains two residual blocks, where the first residual block performs the downsampling task (except encoder1). The decoder has a similar structure to the encoder, except that it adds a transposed convolution at the end for upsampling.

Regarding 3D CPS U-Net, 3D data can be used as input (i.e., a sliding window). During our training and inference stage, each input needs to be resized to 112×160×128 (S×H×W). The 3D network is a 6-layers U-Net, and the encoding process downsamples images to 7×5×4. The residual blocks' structure is similar to 2D CPS U-Net, except that 3D convolutions replace the 2D convolutions.

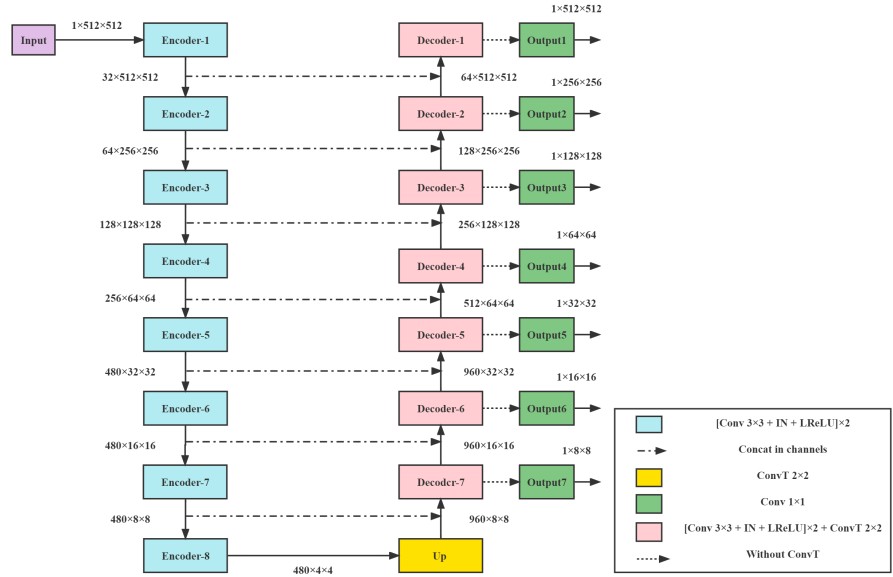

**Fig. 3.** A 2D U-Net like Encoder-Decoder architecture generated by heuristic rules of nnU-Net. The network further entails residual encoder blocks (containing conv-instnorm-ReLU), residual decoder blocks (containing conv-instnorm-ReLU and transpose convolution), and output blocks. The output blocks generate segmentations at different resolutions.

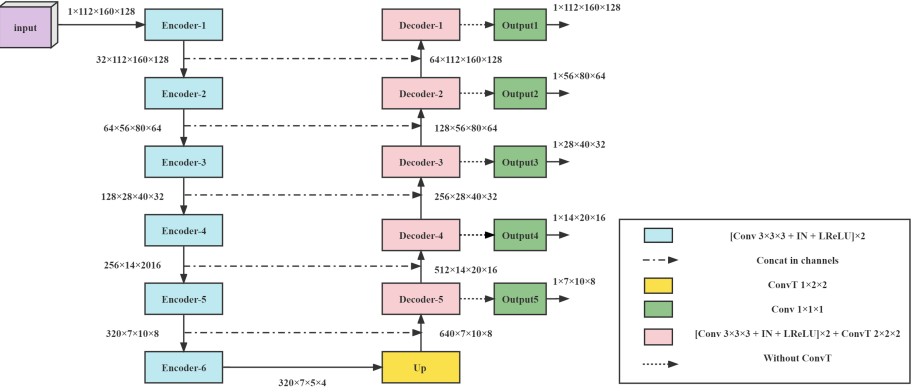

**Fig. 4.** A 3D U-Net like Encoder-Decoder architecture generated by heuristic rules of nnU-Net. The 3D network further entails residual encoder blocks (including conv-instnorm-ReLU), residual decoder blocks (containing conv-instnorm-ReLU and transpose convolution), and output blocks. Same as in the 2D network, the output blocks generate segmentations at different resolutions.

**Inference speed and resources consumption trade-offs** Generally, the performance of the segmentation network is contrary to inference speed and resource consumption. Most methods improve performance at the expense of inference speed or more resource consumption, such as Test Time Augmentation(TTA) inference or sliding window inference.

Due to the limitation of the evaluation platform in the FLARE2022, e.g., the memory limit of 28 GB, we propose enforced spacing settings when making our test submission to reduce resource consumption and speed up the inference time. So, by modifying its spacing parameters before preprocessing, the input patch size is constrained to an optimal value for the inference stage. It is worth mentioning that this strategy is not applied in our experiments except for results on the final test set, which is discussed furthermore in section 4.3.

### 2.3   Post-processing

The spacing of the segmentation output needs to be modified back to its original spacing in the post-processing stage if the enforced spacing settings is used in the preprocessing stage. The remaining post-processing method is consistent with the pipeline of the nnU-Net.

## 3   Experiments

### 3.1   Dataset and evaluation measures

The FLARE 2022 is an extension of the FLARE 2021 [14] with more segmentation targets and more diverse abdomen CT scans. The FLARE 2022 dataset is curated from more than 20 medical groups under the license permission including MSD [15], KiTS [16,17], AbdomenCT-1K [18], and TCIA [19]. The training set includes 50 labeled CT scans with pancreas disease and 2000 unlabelled CT scans with liver, kidney, spleen, or pancreas diseases. The validation set includes 50 CT scans with liver, kidney, spleen, or pancreas diseases. The testing set includes 200 CT scans where 100 cases have liver, kidney, spleen, or pancreas diseases and the other 100 cases have uterine corpus endometrial, urothelial bladder, stomach, sarcomas, or ovarian diseases. All the CT scans only have image information and the center information is not available. The evaluation measures are Dice Similarity Coefficient (DSC) and Normalized Surface Dice (NSD), and three running efficiency measures: running time, area under GPU memory-time curve, and area under CPU utilization-time curve. All measures are used to compute the ranking score. Moreover, the GPU memory consumption has a 2 GB performance. For the FLARE2022 challenge, we use 50 labeled CT scans and the first 1000 unlabeled CT scans.

### 3.2   Environments and requirements

The configured environments and requirements are provided in Table 1.

### 3.3   Training and inference protocols

The protocols are basically calculated by the heuristic rules of nnU-Net, while data-independent parameters are consistent with nnU-Net. The training and inference protocols of our experiments are provided in Table 2.

**Table 1.** Development environments and requirements.

| | |
|---|---|
| Windows/Ubuntu version | Ubuntu 20.04.1 LTS |
| CPU | AMD EPYC 7742 64-Core Processor |
| RAM | 1.8TB |
| GPU (number and type) | 8 NVIDIA A100 (40G) |
| CUDA version | 11.4 |
| Programming language | Python 3.9 |
| Deep learning framework | Pytorch (1.10), nnU-Net |

**Table 2.** Training and Inference protocols.

| Mode | 2D+CPS | 3D+CPS |
|---|---|---|
| Network initialization | "he" normal initialization | "he" normal initialization |
| Batch size | 12 in $D_l$ and 12 in $D_u$ | 2 in $D_l$ and 2 in $D_u$ |
| Patch size | 512×512 | 112×160×128 |
| Total epochs | 1000 | 1000 |
| Optimizer | SGD | SGD |
| Weight decay | 3e-5 | 3e-5 |
| Initial learning rate (lr) | 0.01 | 0.01 |
| Lr scheduler | ReduceLROnPlateau | ReduceLROnPlateau |
| Training time | 47 hours | 97 hours |
| Loss function | Dice and Cross-Entropy | Dice and Cross-Entropy |
| Number of model parameters | 41.29M[3] | 30.79M |
| Number of flops | 65.55G[4] | 585.43G |

## 4   Results and discussions

### 4.1   Quantitative results on validation set

For the ablation study to analyze the effect of unlabeled data and semi-supervised learning, we use 2D and 3D supervised training strategies (previously deployed by nnU-Net) as the baseline in our experiments. Furthermore, our experiments adopt the same preprocessing method and training protocol. In a semi-supervised scheme, we employ two parallel models with the same structure but different initialization weights, while in a supervised scheme, only a single model is required.

Note that the loss function for the semi-supervised scheme is based on Eq. 3, while the experiments in a supervised scheme are performed without $L_{cps}$.

The quantitative results are provided in Table 3, showing the DSC for 13 organs and mean DSC (mDSC) for all classes. The baseline method refers to fully supervised learning on nnU-Net, while Baseline + CPS are the results of 3D-CPS. Table 3 shows baseline+CPS slightly outperforming baseline for both 2D and 3D networks, by +0.0223 and +0.0167 mDSC respectively.

**Table 3.** Quantitative results on 50 cases of validation set.

| Method | 2D | | 3D | |
|---|---|---|---|---|
| | Baseline | Baseline+CPS | Baseline | Baseline+CPS |
| Liver | 0.9623 | 0.9733 | 0.9717 | 0.9745 |
| RK | 0.7699 | 0.8042 | 0.8863 | 0.8851 |
| Spleen | 0.8936 | 0.9202 | 0.9311 | 0.9559 |
| Pancreas | 0.7646 | 0.7865 | 0.8619 | 0.8769 |
| Aorta | 0.9392 | 0.9630 | 0.9584 | 0.9617 |
| IVC | 0.8230 | 0.8545 | 0.8851 | 0.8992 |
| RAG | 0.7124 | 0.7362 | 0.8196 | 0.8354 |
| LAG | 0.6479 | 0.7196 | 0.8060 | 0.8236 |
| Gallbladder | 0.6137 | 0.6993 | 0.7228 | 0.8067 |
| Esophagus | 0.8357 | 0.8461 | 0.8637 | 0.8644 |
| Stomach | 0.8377 | 0.7414 | 0.8800 | 0.9059 |
| Duodenum | 0.6119 | 0.6363 | 0.7586 | 0.7753 |
| LK | 0.7880 | 0.8091 | 0.8699 | 0.8672 |
| mDSC | 0.7846 | 0.8069 | 0.8627 | 0.8794 |

## 4.2   Qualitative results

The qualitative analysis of 20 cases of the validation set released officially is shown in Table 4 and Table 5, including both the score of DSC and NSD. Exemplary segmentation results generated by nnU-Net(baseline) and 3D-CPS (baseline+CPS) are shown in Fig. 5, where case2 and case42 are challenging and case21 and case28 are easy cases. In easy cases, both 2D and 3D networks performed marginally better and improved overall performance. In the challenging cases, the performance of 3D CPS seems to be struggling. Although the overall performance of baseline+CPS is better than the baseline in 20 released cases, the DSC score of baseline+CPS has dropped in these cases(2D model of case2 and 3D model of case42).

**Table 4.** Quantitative analysis of 2D model on 20 cases of validation set.

| Metrics | DSC | | NSD | |
| --- | --- | --- | --- | --- |
| | Baseline | Baseline+CPS | Baseline | Baseline+CPS |
| Liver | 0.9725 | 0.9769 | 0.9487 | 0.9587 |
| RK | 0.7595 | 0.7946 | 0.7225 | 0.7690 |
| Spleen | 0.9374 | 0.9413 | 0.9151 | 0.9355 |
| Pancreas | 0.8043 | 0.8052 | 0.8858 | 0.8944 |
| Aorta | 0.9661 | 0.9717 | 0.9832 | 0.9881 |
| IVC | 0.8403 | 0.8575 | 0.8280 | 0.8615 |
| RAG | 0.7768 | 0.7625 | 0.8782 | 0.8596 |
| LAG | 0.6867 | 0.7817 | 0.7958 | 0.8839 |
| Gallbladder | 0.5034 | 0.6345 | 0.4929 | 0.6328 |
| Esophagus | 0.7690 | 0.8231 | 0.8322 | 0.8980 |
| Stomach | 0.8575 | 0.7217 | 0.8668 | 0.7736 |
| Duodenum | 0.6817 | 0.6580 | 0.8329 | 0.8212 |
| LK | 0.8210 | 0.8495 | 0.7749 | 0.8545 |
| mean | 0.7982 | 0.8137 | 0.8274 | 0.8562 |

**Table 5.** Quantitative analysis of 3D model on 20 cases of validation set.

| Metrics | DSC | | NSD | |
| --- | --- | --- | --- | --- |
| | Baseline | Baseline+CPS | Baseline | Baseline+CPS |
| Liver | 0.9790 | 0.9754 | 0.9736 | 0.9707 |
| RK | 0.8063 | 0.8483 | 0.7927 | 0.8479 |
| Spleen | 0.9436 | 0.9581 | 0.9386 | 0.9434 |
| Pancreas | 0.8737 | 0.8862 | 0.9538 | 0.9589 |
| Aorta | 0.9713 | 0.9730 | 0.9864 | 0.9899 |
| IVC | 0.8897 | 0.8958 | 0.8926 | 0.8937 |
| RAG | 0.8596 | 0.8651 | 0.9568 | 0.9527 |
| LAG | 0.8386 | 0.8521 | 0.9228 | 0.9301 |
| Gallbladder | 0.6626 | 0.7998 | 0.6553 | 0.7946 |
| Esophagus | 0.8752 | 0.8861 | 0.9411 | 0.9438 |
| Stomach | 0.8891 | 0.8869 | 0.9082 | 0.9182 |
| Duodenum | 0.7682 | 0.7727 | 0.8930 | 0.8733 |
| LK | 0.8651 | 0.8537 | 0.8453 | 0.8490 |
| mean | 0.8632 | 0.8810 | 0.8969 | 0.9128 |

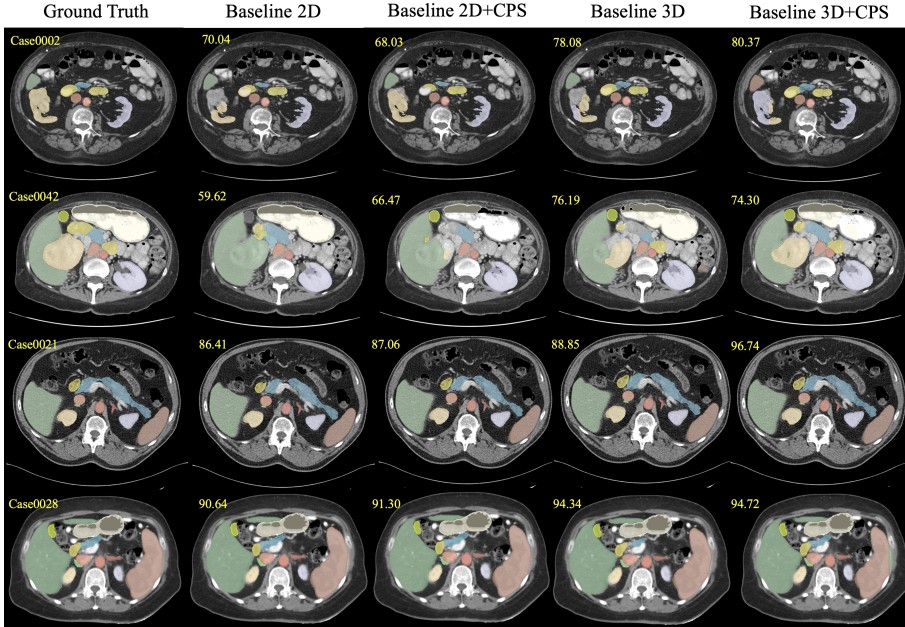

**Fig. 5.** Qualitative results on challenging (case 0002 and 0042) and easy (case 0021 and 0028) cases. Mean DSC scores(%) are attached to the top left. The first column is the ground truth released by the official. The second and fourth columns are the segmentation mask generated from the 2D model and 3D model in nnU-Net, respectively, and the third and fifth columns are generated from 3D-CPS (baseline+CPS).

### 4.3   Results on test dataset

To adapt evaluation platform for our model, we apply the enforced spacing settings strategy in the inference stage. The enforced spacing settings is an empirical parameter strategy that needs to be set manually. For this purpose, we consider the number of slices (S) of CT images for spacing settings, where $s_x$,$s_y$ and $s_z$ represent spacing on the x, y, and z axes, respectively. In our experiments, default settings for $s_x$,$s_y$ and $s_z$ are 0.75, 0.75, 0.5. So, if S is less than 150, only the spacing of $s_x$ and $s_y$ are modified, and $s_z$ will be set as the original spacing. If S is between 150 and 600, the spacing will be modified as the default spacing, and if S is greater than 600, $s_x$ and $s_y$ will be modified as default settings, and $s_z$ will be set to $max(0.8, 600/S) * s_z$.

Other Specific parameters of inference of test submission are set as follows: the TTA inference and post-processing are disabled, whereas the step size of the sliding window is set to 0.7, and the inference mode is set to "fastest mode" as in nnU-Net.

The evaluation of test submission is shown in Table 6. Applying the strategy of enforced spacing settings to the 3D-CPS causes a dramatic drop in DSC and NSD for all types of organs by about ten to twenty percent. Although this

strategy can reduce memory consumption and improve inference speed for the nnU-Net based framework, it proves brute-forced for trade-off accuracy and efficiency, especially for the training set and testing set with different distributions like FLARE 2022 [18].

**Table 6.** Quantitative analysis of test set with the enforced spacing settings.

| Metrics | DSC | NSD |
|---|---|---|
| Liver | 0.8729 | 0.8384 |
| RK | 0.7073 | 0.7074 |
| Spleen | 0.7366 | 0.7345 |
| Pancreas | 0.6301 | 0.7018 |
| Aorta | 0.8200 | 0.8316 |
| IVC | 0.8135 | 0.8115 |
| RAG | 0.6587 | 0.7413 |
| LAG | 0.6586 | 0.7031 |
| Gallbladder | 0.6153 | 0.6143 |
| Esophagus | 0.6063 | 0.6855 |
| Stomach | 0.6147 | 0.6299 |
| Duodenum | 0.5584 | 0.6842 |
| LK | 0.7341 | 0.7433 |
| mean | 0.6920 | 0.7251 |

## 5   Conclusion

Our anticipated model aims to take part in FLARE 2022 competition. For this purpose, a co-training-based semi-supervised method is developed. The proposed method incorporated a semi-supervised architecture based on Cross-Pseudo Supervision (CPS) and nnU-Net, extending to 2D and 3D medical images for segmentation tasks. The proposed method yields better quantitative results with marginally better qualitative predictions than the baseline(nnU-Net). Moreover, introducing enforced spacing settings in the inference stage leads to low-memory consumption and faster inference. It makes nnU-Net based model available with the limitation of the evaluation platform.

Our future work aims two folds. As a first instance, generating more robust pseudo-labels for semi-supervised learning. As our proposed method's generated pseudo-labels are weak at some iterations, and due to no filter module for these low-quality pseudo-labels, our proposed model's accuracy can be affected for some organs. To address this, morphological methods, such as level set representation and edge detection by generating patch-level confidence scores rather than image-level scores, may potentially remedy this anomaly. Secondly, we plan to optimize resource consumption and improve the efficiency of the nnU-Net-based

framework in the inference stage. So, a better method that can trade off accuracy and efficiency is needed, e.g., a two-stage framework with a coarse-to-fine network proposed in the top-10 works of the FLARE2021 challenge.

**Acknowledgements**  We would like to thank the School-Enterprise Graduate Student Cooperation Fund of Shenzhen Technology University. The authors of this paper declare that the segmentation method they implemented for participation in the FLARE 2022 challenge has not used any pre-trained models nor additional datasets other than those provided by the organizers.

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
