# OpenReview forum: "3D Cross-Pseudo Supervision (3D-CPS): A semi-supervised nnU-Net architecture for abdominal organ segmentation"
_MICCAI.org/2022/Challenge/FLARE_

### Official Review · Reviewer_iicf · 2022-09-12
**This is a well-written piece in reviewer's opinion with extensive experiments and assessment.**

**Rating:** 7
**Confidence:** 3

**Review:**

Authors adopt Cross Pseudo Supervision (CPS) for nnUnet, while also propose forced spacing setting to reduce resources consimption.

This is a well-written piece in reviewer's opinion. The tables and figures are logically structured and easy to understand. Qualitative and quantitative results are provided extensively with subjective comments, which is really informative. The preprocessing, postprocessing configuration in their experiments are clearly described.

Suggest improvements:
- Figure 2, backward arrow should be consistent between supervised and unsupervised branch

---

> ### Author Response · Authors · 2022-10-11
> **Thanks for your positive comments and valuable suggestions to improve the quality of our manuscript**
>
> We have unified the backward arrow in Fig. 2. The backward arrow of our model should be from gt (pseudo-gt) to the model's output. So, the head of supervised backward arrows has been changed to point to T1(2) rather than Y1(2).

---

### Official Review · Reviewer_v5d6 · 2022-09-14
**3D Cross Pseudo Supervision (3D-CPS)**

**Rating:** 7
**Confidence:** 4

**Review:**

Summary:

In this work, the authors propose a new 3D Cross Pseudo Supervision (3D-CPS) method, where the nnU-net architecture first modified for semi-supervised tasks.

Strengths:

The labeled data and unlabeled data are calculated based on cross pseudo supervision during the training process to deal with the imbalance problem in the number of labeled data and unlabeled data.
A force spacing settings strategy is proposed to reduce resources consumption and speed up the inference time.

Weaknesses:

No proper analysis of results on the decrease in the metrics of some organs occurred after the use of additional semi-supervised modules.

Details:

1. The English of your manuscript must be improved before resubmission. Many sentences contain grammatical and/or spelling mistakes or are not complete sentences.
2. The characters in Figure 2, Figure 3, and Figure 4 are very small, please enlarge them appropriately.
3. The strategy of uniform slice spacing is a relatively common pre-processing operation. This approach can't be a contribution.
4. In Table 4 and Table 5, the decrease in the metrics of some organs occurred after the use of additional semi-supervised modules, and it is recommended that the authors appropriately discuss the reasons for the occurrence of this phenomenon.
5. In Section 4.3, considering that this method uses the "enforced spacing settings" strategy and the network input is the resized date, why the sliding window is still needed here?

---

> ### Author Response · Authors · 2022-10-11
> **Thanks for your positive comments and valuable suggestions to improve the quality of our manuscript**
>
> 1. Thank you for pointing this out. We have checked the grammatical/spelling mistakes and incomplete sentences accordingly and tried our best to polish the language in the revised manuscript.
> 2. The figures of our paper are already the maximum size under the premise of a fixed layout. We will submit high-resolution figures in the supplemental materials.
> 3. In fact, the enforced spacing strategy is slightly different from uniforming slice spacing in the pre-processing stage. The enforced spacing strategy is to modify the spacing of CT images before pre-processing, while uniforming slice spacing is to unify the spacing of whole CT images from the dataset. Originally, we proposed this strategy to make our model available on the evaluation platform, since the nnunet-based model will occur memory overflow with the limitation of 28GB. However, it proves that this strategy will cause a dramatic drop according to the result evaluated in the final test set. So, it is deprecated and only used in the final test set.
> 4. To some extent, we think this may be due to no relative module for filtering low-quality pseudo-labels. During the training stage, the pair of pseudo-labels generated by the CPS may be poor-quality at some iterations, while it is agnostic for optimal local search of these two branches. Generally, low-quality pseudo-labels do not appear for some easy-predict organs. By contrast, in some cases/organs with a disease, CPS may generate low-quality pseudo-labels, which causes a decrease in the accuracy of some organs.
> 5. Modifying spacing using the enforced spacing settings strategy will reduce the resolution after resampling, but the original image cannot be directly resized to the same resolution as the patch size. Its resolution is still greater than the patch size, so the sliding window is necessary.
>
> In addition, we have added more discussion to elaborate on Question 3&4 in section 4.2.

---

### Official Review · Reviewer_Fcb6 · 2022-09-18
**A semi-supervised nnU-Net architecture for abdominal organ segmentation**

**Rating:** 7
**Confidence:** 3

**Review:**

The article uses the nn-UNet-based framework combined with CPS (cross-pseudo-supervised) as the main method. nn-UNet has achieved good results in many medical image segmentation tasks, and it is a good choice to improve on this basis. Similar to the original author of UNet, the author does not focus on improving the network structure. The basic structure of its network is still the base UNet, but a specific preprocessing method is used for specific tasks to calculate the intensity distribution of the data. Secondly, the composite loss function which has been proved is an effective strategy for improving segmentation results. But there are still some areas to be improved:
1. For preprocessing methods different from nn-UNet, there could be a more detailed description
2. Secondly, there is a lack of relevant explanations on whether forced spacing settings strategy in the inference stage to speed up the inference time will cause changes in the segmentation results.
3. Finally, there can be a general description of the possible improvement directions

---

> ### Author Response · Authors · 2022-10-11
> **Thanks for your positive comments and valuable suggestions to improve the quality of our manuscript**
>
> 1. As we mentioned in section 2.1, the difference between our work and nnU-Net has been described in detail:
>
>    > We use the same pre-processing pipeline generated from the heuristic rules as in the nnU-Net on labeled data, including intensity transformation, spatial transformation, and data augmentation. The only difference is that when analyzing the distribution of CT intensity, nnU-Net analyzes the intensity of the foreground so that the label is required.
>
> 2. In fact, our work is a nnU-Net based method, while the plain nnU-Net cannot be run directly on the evaluation platform for the limitation of the evaluation platform, e.g., the memory limit of 28 GB. To solve the question above, we proposed the enforced spacing strategy to ensure our method works on the evaluation platform. This strategy does reduce memory consumption and speeds up the inference time. However, as concerned, it will cause a dramatic drop according to the result evaluated in the final test set. So, it is deprecated and only used in the final test set. More discussion about this is supplemented in section 4.2.
>
> 3. As suggested, we have added a discussion about future work in section 4.2. Here is the summary from there:
>
>    >The future works can be categorized into two folds:
>    >
>    >1. Generating a more robust pseudo-label for semi-supervised learning.
>    >2. Optimizing resource consumption and improving the efficiency of nnU-Net based framework in the inference stage.

---

### Public Comment · ~Zhengshan_Huang1 · 2022-09-21
**The article is clearly structured and contains all the necessary content.**

The article is clearly structured and contains all the necessary content.

---

> ### Author Response · Authors · 2022-10-11
> **Thanks for your positive comments**
>
> Thanks for your positive comments!

---

### Meta-Review · Program_Chairs · 2022-09-28

**Recommendation:** Major Revision
**Confidence:** 5

**Metareview:**

Reviewers raise many concerns and suggestions. Please address all comments in the revised manuscript.

---

> ### Author Response · Authors · 2022-10-11
> **We have made a point-by-point response to the reviewers' comments and revised our manuscript.**
>
> We have made a point-by-point response to the reviewers' comments and revised our manuscript.